# Transcription, Maturation and Degradation of Mitochondrial RNA: Implications for Innate Immune Response

**DOI:** 10.3390/biom15101379

**Published:** 2025-09-28

**Authors:** Chaojun Yan, Jianglong Yu, Hao Lyu, Shuai Xiao, Dong Guo, Qi Zhang, Rui Zhang, Jingfeng Tang, Zhiyin Song, Cefan Zhou

**Affiliations:** 1National “111” Center for Cellular Regulation and Molecular Pharmaceutics, Key Laboratory of Fermentation Engineering (Ministry of Education), Cooperative Innovation Center of Industrial Fermentation (Ministry of Education & Hubei Province), Hubei Key Laboratory of Industrial Microbiology, Hubei University of Technology, Wuhan 430068, China; ycj114@whu.edu.cn (C.Y.); haolyu@hbut.edu.cn (H.L.); xiaoshuai825@hbut.edu.cn (S.X.); jk1103@whu.edu.cn (D.G.); zhangqi@hbut.edu.cn (Q.Z.); zhangrui1987@hbut.edu.cn (R.Z.); jingfeng_hut@163.com (J.T.); 2Department of Pathology, School of Basic Medicine, Tongji Medical College, Huazhong University of Science and Technology, Wuhan 430030, China; yujianglong@whu.edu.cn

**Keywords:** mitochondrial RNA, transcription, maturation, degradation, immune responds

## Abstract

Mitochondria are crucial for a wide range of cellular processes. One of the most important is innate immunity regulation. Apart from functioning as a signaling hub in immune reactions, mitochondrial nucleic acids can themselves act as damage-associated molecular patterns (DAMPs) to participate in immune processes directly. This review synthesizes the current understanding of mitochondrial RNA (mtRNA) biology and its link to immune activation through aberrant accumulation. We focus on its origin through bidirectional mitochondrial transcription and metabolism, encompassing maturation (cleavage, polyadenylation, modification) and degradation. Dysregulation of mtRNA metabolism leads to mt-dsRNA (mitochondrial double-stranded RNA) accumulation, which escapes mitochondria via specific channels into the cytosol and serves as DAMPs to trigger an immune response. We discuss the critical roles of key regulatory factors, including PNPT1 (PNPase, Polyribonucleotide Nucleotidyltrans ferase 1), in controlling mt-dsRNA levels and preventing inappropriate immune activation. Finally, we review the implications of mt-dsRNA-driven inflammation in human diseases, including autoimmune disorders, cellular senescence, and viral infection pathologies, highlighting unresolved questions regarding mt-dsRNA release mechanisms.

## 1. Introduction

Mitochondria are ubiquitously distributed and indispensable intracellular organelles in eukaryotic cells. Mitochondria are semi-independent double membrane-bound organelles with their own DNA genome, known as mitochondrial DNA (mtDNA). The human mitochondrial genome is a 16,569 bp double-stranded circular DNA and is packaged by a ranged of proteins [1,2]. These two individual strands are referred to as heavy (H) strands and light (L) strands according to the differentiated guanine (G) content [3,4]. mtDNA encode a total of 37 genes, which include 22 transfer RNAs (tRNAs), 2 ribosomal RNAs (rRNAs), and 13 messenger RNAs (mRNAs) that specify core subunits of the OXPHOS (oxidative phosphorylation) system (Figure 1). Beyond their primary role as the major site of ATP generation through OXPHOS, mitochondria are also central to diverse cellular processes, including cell death pathways (apoptosis, pyroptosis, ferroptosis, necrosis, etc.), signal transduction, cell cycle regulation, cellular metabolism, and immune responses [5,6,7,8].

Mitochondria participate in immune responses in two distinct ways. On the one hand, they serve as a platform for antiviral signaling proteins in innate immune reactions. On the other hand, mitochondrial nucleic acids (mt-NAs) act as damage-associated molecular patterns (DAMPs) to engage in immune responses. Mitochondrial antiviral signaling proteins (MAVS), a key adapter protein in the innate immune response to RNA viruses, are located on the outer mitochondrial membrane. It serves as a signaling hub downstream of RIG-I-like receptors (RLRs). By activating interferon (IFN) and inflammatory cytokine pathways, MAVS coordinates the host’s antiviral defense [9,10,11,12,13]. Mitochondrial nucleic acids, including mitochondrial DNA (mtDNA) and mitochondrial RNA (mtRNA), are recognized as non-self-molecules. They function as immune activators, capable of triggering innate immune responses. When mitochondria are damaged by events like oxidative stress or viral infection, mtDNA can escape through several mechanisms, including BAX/BAK macropores, the mPTP (mitochondrial permeability transition pores) and VDAC (voltage-dependent anion channel), or mitochondrial-derived vesicles (MDVs). This released mtDNA then activates the immune response, including the cGAS–STING signaling pathway and the NLRP3 inflammasome [14,15,16,17,18]. Aberrantly accumulated mitochondrial RNA can also leak through pathways such as BAX/BAK and VDAC1, thereby activating the MDA5 (melanoma differentiation-associated gene 5)-driven antiviral signaling pathway and triggering a type I interferon response [19,20]. The mtDNA-cGAS-STING signal pathway is well documented. In this review, we focus on the mtRNA mediated immune responses and diseases caused by their abnormalities. To understand these processes, it is essential to first examine how mtRNA is processed—including transcription, splicing, modification, and maturation—as well as its degradation. These pathways operate in a coordinated manner, and disruptions at any stage can lead to mtRNA abnormalities such as accumulation, which may ultimately trigger immune activation. We will delineate the key steps in mtRNA biogenesis and turnover, and examine specific defects that contribute to mtRNA-driven inflammatory pathways.

## 2. Mitochondrial Transcription Process

Human somatic cells contain thousands of copies of mitochondrial DNA (mtDNA). Unlike nuclear gene expression, mtDNA expression is regulated through mechanisms controlling both the proportion of mtDNA molecules available for transcription and post-transcriptional processes. Mitochondrial transcription is driven by the mitochondrial RNA polymerase (POLRMT) [4,21]. However, POLRMT cannot initiate transcription without the essential factors TFAM (Transcription Factor A, Mitochondrial) and TFB2M (Transcription Factor B2, Mitochondrial). TFAM, the primary protein component of the mitochondrial nucleoid, crucial for mtDNA stability, plays a vital role in transcription initiation [22,23]. It binds to promoters on both the heavy-strand promoter (HSP) and light-strand promoter (LSP), induces DNA bending, and recruits POLRMT via its N-terminal domain to form a binary complex, thereby facilitating assembly of the transcription initiation complex [24,25]. Then, the TFB2M subsequently joins this complex. Functionally analogous to bacterial σ factors, TFB2M inserts into the active site of POLRMT to facilitate promoter melting [26,27]. The POLRMT-TFAM-TFB2M ternary complex catalyzes the formation of the first phosphodiester bond, generating short RNA transcripts (<10 nucleotides). During this initiation phase, POLRMT’s N-terminal domain maintains contact with the promoter, preventing premature complex dissociation [21]. Upon extension of the RNA to 15–20 nucleotides, steric pressure from the nascent RNA-DNA hybrid duplex displaces TFB2M from the polymerase active site [21]. Finally, the transcription elongation factor (TEFM) binds to the catalytic domain of POLRMT. By preventing premature separation of the RNA template strand, TEFM enhances the polymerase’s processivity by >100-fold, enabling POLRMT to transcribe the full 16 kb mitochondrial genome [27,28]. Mitochondrial transcription is terminated by MTERF1 (Mitochondrial Transcription Termination Factor 1) [29]. MTERF1 specifically recognizes the mitochondrial transcription termination sequence on mtDNA [30]. The crystal structure of human MTERF1 reveals that upon binding double-stranded DNA containing this sequence, the protein unwinds the DNA, flipping out three nucleotides critical for termination activity [30]. These flipped-out nucleotides interact with MTERF1 residues, stabilizing the flipped-out base conformation, which leads to dissociation of the POLRMT from the DNA template, thereby terminating transcription [31,32,33]. The transcription process of mitochondrial RNA is shown in the upper left corner of Figure 1.

## 3. Mitochondrial RNA Maturation

Mitochondrial RNA maturation is a complex and highly regulated process that converts polycistronic transcripts into functional mRNAs, tRNAs, and rRNAs. Unlike nuclear RNA processing, mitochondrial RNA maturation occurs in the mitochondrial matrix and involves multiple enzymatic steps, including endonucleolytic cleavage, post-transcriptional modifications (Figure 1) [34,35].

### 3.1. Cleavage of Polycistronic Transcripts

Mitochondrial RNA transcription proceeds bidirectionally from a single intergenic non-coding region (NCR), producing polycistronic RNAs that encompass almost the entire genome. These transcripts contain clusters of mRNAs, rRNAs, and tRNAs, which are flanked by tRNA sequences [36]. The “tRNA punctuation model” hypothesizes that tRNAs serve as recognition sites for endonucleases, guiding the cleavage of polycistronic RNAs into individual functional RNAs [37]. Endonucleases such as RNase P and RNase Z recognize specific tRNA sequences and structures, cleaving the precursor RNA at tRNA boundaries to release mature tRNAs, mRNAs, and rRNAs [38]. RNase Z (also known as ELAC2 in humans) is a metal-dependent endoribonuclease involved in the 3′-end processing of precursor tRNAs (pre-tRNAs) [39,40]. Its primary function is to remove trailer sequences from the 3′ end of pre-tRNAs, generating a terminus suitable for the subsequent addition of the conserved CCA trinucleotide [40,41]. RNase P is a ribonucleoprotein enzyme responsible for the 5′-end processing of pre-tRNAs. Its primary function is to catalyze the cleavage and removal of the 5′ leader sequence from pre-tRNAs, generating a tRNA molecule with a mature 5′ terminus. RNase P and RNase Z typically function sequentially in the tRNA maturation pathway. Their coordinated actions are essential for the complete processing and release of the mature tRNA molecule.

Mitochondrial RNase P is composed of three proteins: TRMT10C (TRNA Methyltransferase 10C, also called MRPP1), HSPD17B10 (Hydroxysteroid 17-Beta Dehydrogenase 10, also called MRRP2), and PRORP (Protein Only RNase P Catalytic Subunit, also called MRRP3) [38]. PRORP serves as the catalytic subunit of mitochondrial RNase P and belongs to the Protein-only RNase P (PRORP) family [42,43]. It is responsible for catalyzing the cleavage of the 5′ leader sequence of precursor tRNAs (pre-tRNAs), a critical step in tRNA maturation [41,44]. However, PRORP cannot exhibit RNase P activity by itself. Its efficient pre-tRNA cleavage function requires the accessory proteins TRMT10C and HSPD17B10. TRMT10C possesses methyltransferase activity and is involved in tRNA modification, specifically catalyzing the N1-methylation of purine bases within tRNA molecules [44,45]. HSPD17B10 forms a stable complex with TRMT10C [46]. Notably, the methyltransferase activity of TRMT10C is not required for supporting the cleavage activity of PRORP. Instead, the TRMT10C-HSD17B10 complex enhances cleavage efficiency primarily by promoting PRORP’s binding to its substrate [41,46]. While it is established that nuclear RNA modifications are co-transcriptionally incorporated, the temporal relationship between TRMT10C-mediated methylation and RNA cleavage during mitochondrial RNA processing remains unclear.

Some mitochondrial transcripts are not punctuated by tRNAs, such as between the genes *ATP6*/*ATP8* and *COX3*. However, the mechanism by which their primary transcripts are processed into separate, mature mRNAs remains unclear. Recent studies have demonstrated that FAST kinase domain-containing protein 4 (FASTKD4) and FASTKD5 may be related to this process [47]. These findings highlight limitations of the tRNA punctuation model. Moreover, the recent identification of diverse types of mitochondrial non-coding RNAs (mt-ncRNAs), including long non-coding RNAs (lncRNAs), small non-coding RNAs (sncRNAs) and identified circular RNAs (circRNAs), further indicates that the tRNA punctuation model cannot fully account for all cleavage events involved in mitochondrial primary transcript processing or the biogenesis of all mitochondrial RNAs [48,49,50]. The mt-ncRNAs’ biogenesis relies on the mitochondrial transcription machinery, and subsequent processing. Their release from primary transcripts may occur through both punctuation-dependent and independent on “tRNA punctuation model”. For example, the maturation of lncRNAs, including lncCytb, lncND5, and lncND6, requires involvement of the mitochondrial RNase P protein 1 (MRPP1) for processing at both the 5′ and 3′ ends [48,51]. Mitochondrion-encoded circular RNAs (mecciRNAs) are formed through head-to-tail splicing of precursor RNAs [50]. Although their precise biogenic pathways are still under active investigation, they are generally transcribed from mtDNA and cyclized within the mitochondria. The 7S RNA is transcribed from the non-coding D-loop region of mitochondrial DNA [52]. Its precursor is an independently transcribed ~180 bp RNA molecule that matures without requiring additional cleavage; instead, it only undergoes 3′ polyadenylation to form the functional transcript. The mature 7S RNA can bind to POLRMT and promote its homodimerization, thereby inhibiting mitochondrial transcription [52].

### 3.2. Mitochondrial RNA Post-Transcriptional Modifications

#### 3.2.1. mt-mRNA Post-Transcriptional Modification

Post-transcriptional modifications are essential for mitochondrial RNA maturation and biofunction. Mitochondrial RNA modifications, although less numerous than their cytoplasmic counterparts, encompass several types and play crucial roles in precise protein synthesis [53]. The most common is the adenylation of the 3′-end mitochondrial mRNA. Except for MT-ND6, most mitochondrial mRNAs undergo 3′ polyadenylation after cleavage from the primary transcript [54,55]. This process is catalyzed by mitochondrial poly(A) polymerase (mtPAP, also known as PAPD1), a homodimeric enzyme [56,57]. For nuclear-encoded genes, polyadenylation plays a critical role in promoting mRNA stability and protein translation [58]. In contrast, in mitochondrial mRNA, polyadenylation functions not only to enhance mRNA stability but also to promote tRNA maturation [59]. The absence of mtPAP results in the loss of polyadenylation across all assayed mitochondrial transcripts, while oligoadenylation is preserved [60]. Importantly, the impact of this loss on transcript stability is transcript-dependent [61]. The other protein related to polyadenylation is LRPPRC (Leucine-Rich PPR Motif-Containing Protein). LRPPRC stabilizes mitochondrial RNA by binding to it and acting as a molecular chaperone [62]. It forms a complex with SLIRP (SRA Stem-Loop Interacting RNA Binding Protein) to maintain the secondary structures of mitochondrial mRNAs, preventing their degradation and ensuring their stability [63]. Studies demonstrate that loss of SLIRP protein significantly impairs the stability of LRPPRC protein [64]. This destabilization subsequently exerts detrimental effects on mitochondrial mRNA levels [64]. In addition, LRPPRC participates in regulating RNA polyadenylation and translation. A recent study reveals that the LRPPRC-SLIRP protein complex binds to mtRNA to stabilize it and expose initiation sites, bridges and positions the mitochondrial ribosomes, regulates translation initiation/elongation via interacting with mitochondrial translation factors, and ensures translation accuracy and mtRNA homeostasis [65]. Notably, mutations in the LRPPRC gene are conclusively linked to the development of Leigh syndrome (LSFC) [66]. The underlying mechanism involves LRPPRC mutations reducing mitochondrial mRNA levels, which compromises the proper assembly and function of cytochrome c oxidase (COX), ultimately leading to the pathogenesis of Leigh syndrome [66].

In addition to polycistronic cleavage and mRNA polyadenylation, base modifications are also part of the mitochondrial RNA maturation process. To date, over 170 chemical modifications have been identified in RNA, affecting the function of diverse RNAs, including nuclear and mitochondrial RNAs [67]. The most prevalent among these modifications in mitochondrial mRNA are methylation and pseudouridylation [68]. Methylation of mitochondrial mRNA (mt-mRNA) predominantly involves N1-methyladenosine (m^1^A), catalyzed by the mitochondrial methyltransferase complex TRMT61B (TRNA Methyltransferase 61B) and TRMT6/61A (TRNA Methyltransferase 6) [69]. This modification disrupts A-U base pairing and reduces mitochondrial translation efficiency. Pseudouridine (Ψ) modifications are putatively mediated by enzymes such as RPUSD3 (RNA Pseudouridine Synthase D3) and TRUB2 (RNA Pseudouridine Synthase Domain Containing 2), although their specific biological functions in mt-mRNA remain poorly characterized [70]. The m^6^A modification, ubiquitous in nuclear and cytosolic mRNAs, has also been detected in mt-mRNAs [71]. However, the methyltransferase(s) responsible for depositing m^6^A within mitochondria remain unidentified.

#### 3.2.2. mt-tRNA Post-Transcriptional Modification

Among these three RNA types, tRNA exhibits the most abundant base modifications. Following its release from polycistronic RNA by RNase P and RNase Z cleavage, the 3′ end of mt-tRNA is matured through the addition of the universally conserved CCA (cytosine–cytosine–adenine) sequence by tRNA nucleotidyltransferase TRNT1 [45,72]. The CCA tail is crucial for tRNA aminoacylation, ribosomal binding, and protein synthesis [73]. Deficiencies in TRNT1 impair tRNA integrity and disrupt protein homeostasis [73]. Mutations or loss of TRNT1 function causes a multisystemic and clinically heterogeneous disease termed SIFD (sideroblastic anemia with B-cell immunodeficiency, periodic fevers, and developmental delay; SIFD) [74]. Additionally, methylation and pseudouridylation are widespread modifications on mitochondrial tRNAs (mt-tRNAs). Specifically, the adenine or guanine at position 9 is commonly methylated to m^1^A9 or m^1^G9, catalyzed by the TRMT10C-HSPD17B10 complex [41]. These modifications stabilize tRNA structure and function, thereby influencing protein translation [65]. TRMT10C is an S-adenosylmethionine (SAM)-dependent methyltransferase that catalyzes the transfer of a methyl group to the N^1^ position of adenine or guanine at position 9 (A9/G9) in mitochondrial tRNA [75]. In contrast, HSPD17B10 likely acts as a chaperone, stabilizing TRMT10C. Mutations in HSPD17B10 result in loss of TRMT10C and impaired processing of mitochondrial heavy-strand polycistronic transcripts [76]. m^5^C (5-methylcytosine) modification has also been detected at position 48–50 of mt-tRNA, which is added by NSUN2 (NOP2/Sun RNA Methyltransferase 2) [77]. TRMT61B was identified to be the methyltransferase of m^1^A at position 58 of several mt-tRNAs, which enhances the stability of the structure [78]. Pseudouridine (Ψ), one of the most abundant tRNA modifications and often termed the ‘fifth nucleotide’, is catalyzed by enzymes including PUS1 (Pseudouridine Synthase 1), RPUSD3, and TRUB2 (TruB Pseudouridine Synthase Family Member 2) [70,79,80]. PUS1 modifies uridine residues at positions 27 and 28 (U27/U28) in tRNA, whereas the specific modification sites for RPUSD3 and TRUB2 remain incompletely characterized [70,79]. Pseudouridine (Ψ) modifications in mt-tRNA contribute to structural stability and are essential for mitochondrial function [81]. Modifications within the anticodon loop of tRNA, particularly those at positions 34 and 37, are crucial for the fidelity and efficiency of codon-anticodon recognition. Among the diverse modifications found at position 37, m^5^C is relatively common. This methylation is catalyzed by the enzyme NSUN3 (NOP2/Sun RNA Methyltransferase 3), and can be reversed by the demethylase ALKBH1 (AlkB Homolog 1) [82,83]. Position 37 exhibits a remarkable variety of modification types. Key examples include m^1^G (1-methylguanosine), added by TRMT5 (TRNA Methyltransferase 5) [84]; i^6^A (N^6^-(dimethylallyl) adenosine), synthesized by TRIT1 (TRNA Isopentenyltransferase 1) [85]; and t^6^A (N^6^-threonylcarbamoyladenosine), whose formation requires the co-catalytic activity of YRDC (YrdC N6-Threonylcarbamoyltransferase Domain Containing) and OSGEPL1 (O-Sialoglycoprotein Endopeptidase Like 1) [86]. Collectively, these modifications at position 37 play a vital role in stabilizing the interaction between the codon and anticodon during translation.

#### 3.2.3. mt-rRNA Post-Transcriptional Modification

Human mitochondrial rRNA, composed of the 16S and 12S large subunits, features three key modification types: 2′-O-ribose methylation, base methylation, and pseudouridylation. In the 16S rRNA, three specific 2′-O-ribose methylation sites exist (Gm1145, Um1369, and Gm1370) catalyzed by MRM1 (Mitochondrial RRNA Methyltransferase 1), MRM2 (Mitochondrial RRNA Methyltransferase 2), and MRM3 (Mitochondrial RRNA Methyltransferase 3), respectively; these modifications are essential for assembling the mitochondrial large ribosomal subunit and enabling mitochondrial protein translation [87,88]. Base methylation, specifically m^1^A, also occurs at position 947 and is catalyzed by TRMT61B [89]. Pseudouridylation at position 1397, mediated by RPUSD4 (RNA Pseudouridine Synthase D4), contributes to large subunit stability and promotes mitochondrial protein translation [70,90].

The 12S rRNA harbors abundant m^4^C and m^5^C base methylations. METTL15 (Methyltransferase 15) catalyzes m^4^C modification at position 839, NSUN4 (NOP2/Sun RNA Methyltransferase 4) mediates m^5^C modification at position 84, and METTL17 (Methyltransferase Like 17) facilitates both m^4^C modification at position 840 and m^5^C modification at position 842 [91,92,93]. The clustered methylations at positions 839–842 facilitate the translation of mitochondrial transcripts [91,92,93]. Loss of METTL17 significantly impairs mitochondrial protein translation and inhibits the mitochondrial oxidative respiratory chain; notably, METTL17 deficiency has also been reported to suppress tumor proliferation [93,94,95]. Similarly, NSUN4 deficiency in mice inhibits mitochondrial translation, while the absence of METTL15 disrupts mitochondrial ribosome assembly [92,96]. 12S rRNA also undergoes m^6^_2_A dimethylation at positions 936 and 937, which is deposited by TFB1M (Transcription Factor B1, Mitochondrial) [97]. The m^6^_2_A dimethylation modification promotes the binding of Ribosome Factor A (RBFA) to rRNA and is essential for ribosome assembly [98].

## 4. Mitochondrial RNA Degradation

Mitochondrial mRNAs typically have a half-life of 1–2 h [99]. The primary complex responsible for their degradation is the hSUV3/PNPase complex, within specialized D-foci in the mitochondrial matrix [99]. PNPase (also named PNPT1) provides 3′ → 5′ exoribonuclease activity, while the ATP-dependent helicase hSUV3 (Suv3 Like RNA Helicase) unwinds structured RNA substrates [99]. This complex preferentially degrades aberrant antisense mtRNAs and contributes to mRNA turnover, generating short oligonucleotides [99,100]. These products are subsequently processed to monoribonucleotides by the 3′ → 5′ exonuclease REXO2 (RNA Exonuclease 2) ensuring nucleotide recycling and preventing accumulation of inhibitory fragments (Figure 1) [101]. Degradation is dynamically opposed by the LRPPRC/SLIRP stabilization complex: LRPPRC and SLIRP cooperatively bind mt-mRNAs/rRNAs, suppress degradosome activity, prevent destructive secondary structures, and facilitate polyadenylation by mtPAP [102]. Critically, mtRNA stability is further enhanced through mitoribosome association; ribosomal binding physically shields transcripts from degradation, with depletion experiments confirming that unbound mRNAs are vulnerable to decay [103].

The degradosome also governs mitochondrial nucleic acid homeostasis beyond canonical decay. By degrading antisense transcripts, it limits mitochondrial double-stranded RNA (mt-dsRNA) accumulation. PNPase’s sublocalization dictates distinct roles: in the matrix (with SUV3), it degrades dsRNA, while in the intermembrane space (SUV3-independent), it sequesters dsRNA, preventing cytosolic leakage and inflammatory responses [104]. The RNA-binding protein GRSF1 (G-Rich RNA Sequence Binding Factor 1) enhances degradation specificity by resolving stable G-quadruplex (G4) structures in antisense RNAs via its quasi-RRM domains, enabling their unwinding and subsequent degradosome targeting—an adaptation to vertebrate G4-rich mitochondrial genomes. Dysregulation of this finely tuned system underlies severe pathologies. PNPase mutations (e.g., Gln387Arg, Glu475Gly) disrupt trimerization, impairing RNA import and causing early-onset neurological disorders (encephalopathy, hearing loss) [105,106]. LRPPRC mutations cause Leigh Syndrome French Canadian type (LSFC), a disorder characterized by the features described above. Notably, LRPPRC overexpression in diverse cancers promotes tumorigenesis by inhibiting apoptosis, enhancing invasion, and suppressing mitophagy through aberrant Parkin interaction [107,108]. Thus, precise mtRNA degradation is indispensable for mitochondrial integrity and cellular health, with defects impacting development, neurology, and oncology.

## 5. Mitochondrial RNA-Mediated Immunity

### 5.1. MAVS Signal Pathway

Mitochondria serve as signaling hubs for immune responses, among which the RNA-mediated MAVS signaling pathway is the most prominent. The MAVS signaling pathway is a pivotal innate immune defense mechanism that orchestrates the host response against viral infections. This intricate cascade is initiated upon the recognition of viral RNA by specialized pattern recognition receptors (PRRs) localized in various cellular compartments, including the cytoplasm and endosomes [109]. Key PRRs involved in this process include the retinoic acid-inducible gene I (RIG-I)-like receptors (RLRs), such as RIG-I and melanoma differentiation-associated gene 5 (MDA5), as well as toll-like receptors (TLRs) like TLR3, TLR7, and TLR8 [110,111]. RIG-I selectively binds to short double-stranded RNA (dsRNA) with 5′ triphosphate ends, a molecular signature often presents in viral genomes or replication intermediates [112,113]. In contrast, MDA5 recognizes long dsRNA molecules, which are typical byproducts of viral replication [114]. TLR3, localized in endosomes, detects dsRNA, while TLR7 and TLR8 (expressed in immune cells like plasmacytoid dendritic cells) sense single-stranded RNA (ssRNA) from viruses [115,116].

Upon binding to viral RNA, these PRRs undergo conformational changes and activate downstream signaling. For RIG-I and MDA5, this activation involves the recruitment of the adaptor protein-MAVS, which is anchored to the mitochondrial outer membrane [11]. The interaction between activated RLRs and MAVS is mediated by homophilic caspase recruitment domain (CARD) interactions: RIG-I and MDA5 possess N-terminal CARDs that bind to the CARD of MAVS, triggering MAVS oligomerization [117,118].

MAVS oligomerization is a critical step that converts the initial RNA recognition event into a robust signaling response. Oligomerized MAVS forms large signaling complexes on the mitochondrial membrane, often referred to as “MAVS aggregates,” which serve as a platform for recruiting downstream signaling molecules [117]. These aggregates propagate the signal through two major pathways: the NF-κB (Nuclear Factor Kappa B Subunit 1)pathway and the interferon regulatory factor (IRF) pathway [117].

In the NF-κB pathway, MAVS recruits tumor necrosis factor receptor-associated factors (TRAFs), such as TRAF2, TRAF3, and TRAF6 [119,120,121]. TRAF6 activates the TAK1 (transforming growth factor-β-activated kinase 1) complex, which in turn phosphorylates IκB kinase (IKK) subunits [122]. Phosphorylation of IκB (inhibitor of NF-κB) leads to its ubiquitination and degradation, releasing NF-κB to translocate into the nucleus and induce the expression of pro-inflammatory cytokines, such as tumor necrosis factor-α (TNF-α) and interleukin-6 (IL-6) [123].

For the IRF pathway, TRAF3 recruited to MAVS activates the TBK1 (TANK-binding kinase 1) and IKKε (IKK-epsilon) kinases. These kinases phosphorylate IRF3 and IRF7, transcription factors that then dimerize and translocate to the nucleus. Activated IRF3 and IRF7 drive the expression of type I interferons (IFNs), primarily IFN-α and IFN-β, which are key antiviral cytokines [124].

Another pathway activated by mitochondrial RNA involves PKR-mediated signaling. PKR (double-stranded RNA-dependent protein kinase) is a serine/threonine protein kinase initially identified as an interferon (IFN)-inducible, innate immune antiviral protein activated by dsRNA [125]. Kim et al. demonstrated that PKR senses not only viral dsRNA but also endogenous nuclear and mitochondrial dsRNA [126]. The binding of dsRNA to PKR’s dsRNA-binding domains (dsRBDs) promotes dimerization and autophosphorylation, thereby activating its kinase function [127]. Activated PKR can trigger the NF-κB signaling pathway, leading to the expression of pro-inflammatory cytokines [128]. Additionally, PKR phosphorylates the eukaryotic translation initiation factor 2α (eIF2α), inhibiting global protein synthesis to restrict viral replication [128]. Under specific conditions, PKR activation can also induce apoptosis [129].

### 5.2. Abnormal Mitochondrial dsRNA Accumulation Induces Immune Responses

The mitochondrial genome undergoes bidirectional transcription, generating overlapping transcripts that can form long double-stranded RNA structures [19]. Normally, the levels of mt-dsRNA are tightly regulated by the mitochondrial RNA degradosome components, such as mitochondrial RNA helicase SUV3 (hSUV3) and polynucleotide phosphorylase PNPase (PNPT1), along with its cofactor GRSF1, which have been described above [100]. However, in various pathophysiological states, this processing can be disrupted, leading to the abnormal accumulation of mtRNA. Proudfoot’s group identified *PNPT1* mutation in blood leukocytes of patients. *PNPT1* mutations impair PNPase function, reducing mt-dsRNA degradation and causing its accumulation. This mt-dsRNA escapes into the cytoplasm via Bax-Bak pores and is sensed by MDA5, activating the MAVS-mediated signaling pathway to trigger a type I interferon response, with elevated interferon-stimulated genes (ISGs) and markers of immune activation (Figure 2) [100]. Although SUV3 deficiency causes mt-dsRNA accumulation, it fails to trigger an immune response, even when ABT-737 (Bcl2 and Bcl-xL inhibitor) is administered [100]. The underlying mechanism for this remains unclear. Mitochondrial mRNA (mt-mRNA) is polyadenylated by mtPAP and stabilized by the LRPPRC-SLIRP complex. Studies in *Drosophila* revealed that, in addition to PNPase deficiency, loss of mtPAP function also promotes mt-mRNA accumulation and activates immune signaling in flies [130]. These findings indicate that aberrant accumulation of mt-dsRNA is a key factor triggering mt-RNA-mediated immune responses. Leveraging this characteristic, Sujin Kim and colleagues employed CRISPR-CAS9 screening to identify 89 mitochondrial RNA-binding proteins which, when lost, cause mt-dsRNA accumulation. Notably, they found that loss of the rRNA methyltransferase NSUN4 (NOP2/Sun RNA methyltransferase 4) promotes mt-dsRNA accumulation and immune activation through activating PKR signal pathway. Mechanistically, NSUN4 catalyzes the formation of m^5^C-modified RNA, which is recognized by C1q-binding protein (C1QBP). C1QBP subsequently recruits PNPT1 to the RNA for degradation, indicating that m^5^C marked mtRNA for degradation (Figure 2) [131].

Metabolites actively regulate mitochondrial RNA (mtRNA)-mediated immune responses. As demonstrated by Hooftman et al., LPS stimulation of macrophages induces metabolic rewiring, including an aspartate-argininosuccinate shunt driven by increased ASS1 (Argininosuccinate Synthase 1) expression [132]. This elevates cytosolic fumarate, promoting protein succination [132]. Critically, pharmacological or genetic inhibition of fumarate hydratase (FH) further increases fumarate, suppresses mitochondrial respiration, and elevates mitochondrial membrane potential. These changes reduce IL-10 and increase TNF secretion (mimicked by fumarate esters). Notably, FH inhibition uniquely triggers mtRNA release, activating RNA sensors (TLR7, RIG-I, MDA5) to induce IFN-β production, an effect also seen with endogenous FH suppression after prolonged LPS. FH suppression is observed in systemic lupus erythematosus (SLE) patient cells, suggesting its pathogenic role, indicating FH acts as a protective regulator, restraining mtRNA release to maintain appropriate cytokine and interferon responses and prevent aberrant immune activation [132].

Except for systemic lupus erythematosus (SLE), other autoimmune diseases such as Sjögren’s syndrome have also been found to have mt-dsRNA-mediated immune activation [133]. A recent study showed that the release of mt-dsRNA into the cytosol acts as a critical factor in inducing and maintaining the inflammatory phenotype of senescent cells, which is similar to mtDNA release [134,135]. Furthermore, mt-dsRNA release might be connected to viral infections and heart diseases. Recent studies have reported EMCV (encephalomyocarditis virus) infection disrupts normal mitochondrial functions, which may contribute to the release of mitochondrial dsRNA [19]. Additionally, EMCV-induced mitochondrial dysfunction and mt-dsRNA release could contribute to the tissue damage seen in infected organisms. In cases of myocarditis caused by EMCV, the leakage of mt-dsRNA into the extracellular space might recruit immune cells, exacerbating inflammation in the heart muscle. However, the potential mechanisms underlying the interplay between EMCV infection and mt-dsRNA release remain largely unelucidated and require in-depth investigation.

### 5.3. Mitochondrial Non-Coding RNAs Mediated Immune Responses

Ashish Dhir et al. used RNA sequencing profile to demonstrate that released mitochondrial double-stranded RNA (mt-dsRNA) covers the entire mitochondrial genome [19]. We hypothesize that mitochondrial non-coding RNAs may also act as damage-associated molecular patterns (DAMPs) capable of activating immune responses. Although there is no direct evidence supporting this hypothesis, it is plausible that mt-ncRNAs participate indirectly in immune regulation by modulating mitochondrial functions—such as oxidative phosphorylation and reactive oxygen species (ROS) production—which in turn influence inflammatory pathways.

lncFAO (fatty acid oxidation-associated lncRNA), which is nuclear-encoded mitochondria-located lncRNAs, is identified in mouse macrophages at a later time after LPS stimulation. lncFAO binds to HADHB (a subunit of the mitochondrial trifunctional enzyme, critical for β-oxidation) and increases HADHB protein levels to promote fatty acid oxidation (FAO) [136]. Fatty acid oxidation is essential for M2 macrophage polarization. By enhancing FAO, lncFAO supports M2-mediated resolution of inflammation [136].

SCAR/mc-COX2 (Steatohepatitis-associated circRNA ATP5B regulator/mt-circ-COX2), which is produced from the COX2 locus on the L-strand, is downregulated in liver fibroblasts from patients with nonalcoholic steatohepatitis (NASH) and upregulated in plasma exosomes of chronic lymphocytic leukemia (CLL) patients [137]. SCAR binds to ATP5B and inhibits the interaction between cyclophilin D (CypD, an activator of the mitochondrial permeability transition pore) and the mPTP complex. This prevents the opening of the pore and reduces the release of mitochondrial reactive oxygen species (mtROS). Deficiency in SCAR leads to accumulation of mtROS, which promotes the activation of liver fibroblasts and exacerbates metaflammation. Notably, mitochondrial delivery of SCAR using mito-NP nanoparticles has been shown to ameliorate NASH in mouse models [137].

## 6. Future Perspectives

Current research reveals that mt-dsRNA accumulation is a primary factor triggering RNA-mediated inflammatory responses. This aberrant RNA accumulation stems from deficiencies in the PNPT1-SUV3 complex or REXO2 function [101]. Interestingly, however, accumulation caused by SUV3 deficiency alone does not induce inflammation [19]. PNPT1 requires SUV3 as a cofactor for its RNA degradation activity and is catalytically inert when alone. PNPT1 is primarily localized to the mitochondrial matrix, although it has also been reported to degrade RNA in the intermembrane space (IMS) and cytosol [138]. Since SUV3 is exclusively expressed in the matrix, this prompts the following question: does another protein facilitate PNPT1-mediated RNA degradation within the IMS? If this protein exists, it would offer a compelling explanation for why mtRNA accumulation caused by loss of SUV3 does not induce an immune response—namely, PNPT1 may clear such RNA within the IMS.

Sujin Kim et al. reported that m^5^C-marked mt-rRNAs are recognized by C1QBP and subsequently degraded by PNPT1 [131]. In contrast, loss of the m^5^C methyltransferase NSUN4 promotes dsRNA accumulation and immune activation. This suggests that RNAs are marked for degradation. A recent study showed that pseudouridine (Ψ)-containing RNA evades immune detection by Toll-like receptors (TLR7 and TLR8) [139], indicating that RNA modification is a marker for immunity recognition. A series of key questions has emerged from these findings. Firstly, what are the specific molecular marks of RNA that play a crucial role in promoting RNA release? Do these released mt-dsRNAs carry distinct molecular signatures that promote their selective export?

Secondly, which proteins, apart from the ones already mentioned, are involved in this RNA release process? And finally, what are the underlying molecular mechanisms that govern the entire sequence of events related to RNA release? RNA typically functions in conjunction with RNA-binding proteins (RBPs). In addition to RNA modifications such as m^5^C and pseudouridine, which were mentioned earlier, the selective clearance or release of RNA may also be regulated by specific RBPs. These proteins could act as receptor-like molecules or adaptors that promote RNA degradation or export. So far, however, C1QBP (Complement Component 1 Q Subcomponent- Binding Protein) remains the only protein known to play such a role. We propose that other proteins with similar functions likely exist. As demonstrated by Sujin Kim et al., RNA immunoprecipitation (RIP) could serve as an effective method to isolate and identify such candidate RBPs. All these aspects are areas that demand further in-depth investigation to gain a more comprehensive understanding of the RNA-related processes in question.

How does mt-dsRNA leak from mitochondria? The mechanisms underlying mtDNA release are relatively well-characterized: mtDNA can be released into the cytosol via BAX/BAK pores, or through VDAC, the mPTP, GSDMD, and, more recently reported, p-MLKL (Mixed Lineage Kinase Domain Like Pseudokinase) pores [140,141,142,143]. Similarly, mt-dsRNA release can also occur via BAX/BAK pores and VDAC, with ANT2 recently identified as an additional released channel [19,144]. Notably, mtDNA can also be released through mitochondrial-derived vesicles (MDVs) or via fusion with multivesicular bodies (MVBs) for extracellular export [145]. Therebefore, we hypothesize that mtRNA may be released through additional mechanisms. For example, accumulated mtRNA might escape mitochondria through pores formed by MLKL or other mediators such as BSDMD, potentially triggering immune activation. Alternatively, mtRNA could be packaged into mitochondrial-derived vesicles (MDVs) and subsequently delivered to endosomes, where it engages Toll-like receptor 3 (TLR3) to initiate an immune response (Figure 2). These requires further investigation.

## Figures and Tables

**Figure 1 biomolecules-15-01379-f001:**
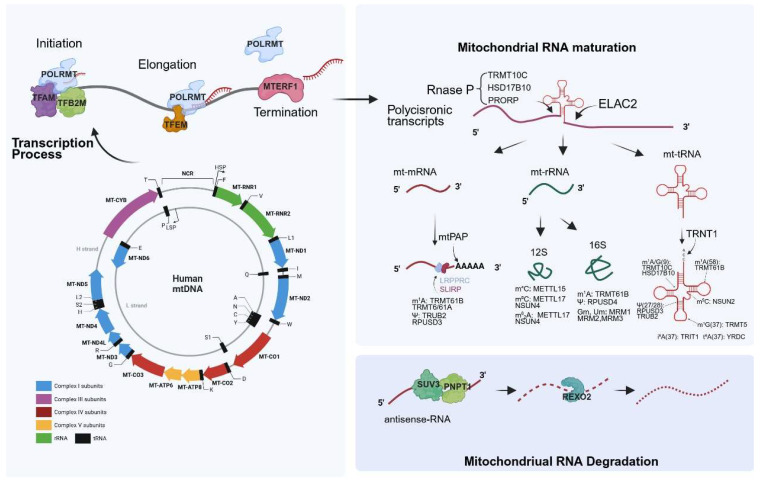
Schematic representation of human mitochondrial DNA (mtDNA) and fundamental stages of mitochondrial RNA (mtRNA) metabolism. The mitochondrial genome encodes 37 genes, including 22 transfer RNAs (tRNAs), 2 ribosomal RNAs (rRNAs), and 13 messenger RNAs (mRNAs). Mitochondrial DNA transcription is initiated by POLRMT with the essential assistance of TFAM and TFB2M. This process generates bidirectional polycistronic transcripts from both the heavy and light strands. These primary transcripts are subsequently processed by RNase P and RNase Z (ELAC2), which cleave at tRNA boundaries to release individual mRNAs, tRNAs, and rRNAs. Following cleavage, the liberated RNAs undergo diverse post-transcriptional modifications catalyzed by specific enzymes. These modifications serve critical functions, including stabilizing RNA structure and facilitating translation. Finally, mitochondrial RNA degradation is mediated by the PNPT1-SUV3 complex and REXO2. The solid line refers to antisense-RNA. The middle dashed line represents the small RNA fragments degraded by PNPT1, and the rightmost dashed line represents the RNA degraded by REXO2. Created in BioRender. Created in BioRender. Yan, C. (2025) https://BioRender.com/a4gafh7.

**Figure 2 biomolecules-15-01379-f002:**
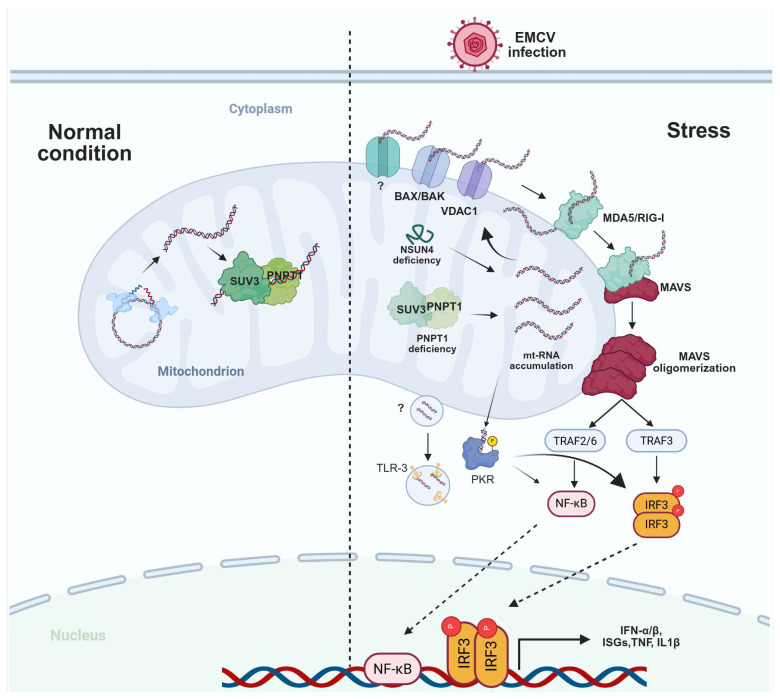
Released mtRNA can activate immune responses. Under normal conditions, mt-dsRNA generated within mitochondria is degraded by the PNPT1-SUV3 complex and subsequently processed by REXO2. However, under stress conditions—such as PNPT1 inactivation or NSUN4 deficiency—mt-dsRNA accumulates abnormally. This accumulated mt-dsRNA escapes the mitochondria into the cytosol via channels in the outer mitochondrial membrane, including BAX/BAK, VDAC1, and potentially other unidentified channels or mechanisms like mitochondrial-derived vesicles (MDVs). In the cytosol, the released mt-dsRNA is recognized by sensors including MDA5, RIG-I, PKR, and TLR3. This recognition triggers immune signaling pathways, leading to the production of type I interferons and other inflammatory mediators. All line segments with arrows indicate promotion. The long dashed line represents the dividing line between normal conditions and stress conditions. Created in BioRender. Yan, C. (2025) https://BioRender.com/yb3vr4x.

## Data Availability

No new data were created or analyzed in this study. Data sharing is not applicable to this article.

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
