# Peer review of "Transcription, Maturation and Degradation of Mitochondrial RNA: Implications for Innate Immune Response"

_biomolecules, 2025, doi:10.3390/biom15101379_

Round 1
Reviewer 1 Report
Comments and Suggestions for Authors
The article “Transcription, maturation and degradation of mitochondrial RNA: implications for innate immune response” by Yan et al. provides a thorough review of mitochondrial RNA metabolism machinery. I enjoyed reading the manuscript. Below are some of my comments and suggestions to improve the article (these are examples):
- Page 1, line 37: Change double-strander to double-stranded. Randed to range.
- Page 1: Provide a citation to direct readers interested in the mtDNA interactome for the sentence: “The human mitochondrial genome is a 16,569 bp double-strander circular DNA and is packaged by a ranged of proteins.”
- Page 1, line 39: Replace distribution with content.
- Page 2, line 54: Provide a more appropriate citation for MAVS to direct interested readers.
- Page 2, line 59: Rewrite the sentence:
“Upon mitochondrial are damaged, such as oxidative stress or viral invasion, can lead to the release of mtDNA via pathways like BAX/BAK macropores, mPTP channel, and mitochondrial-derived vesicles (MDVs), thereby activating the cGAS-STING signaling pathway.”
for better clarity and grammar. - Page 2, line 64: Replace citation #14 with PMID: 39406969, as the cited article does not discuss VDAC1 in the release of mtRNAs.
- Page 2, line 92: Define MTERF1.
Overall: The article is well-structured and easy to read.
Author Response
The article “Transcription, maturation and degradation of mitochondrial RNA: implications for innate immune response” by Yan et al. provides a thorough review of mitochondrial RNA metabolism machinery. I enjoyed reading the manuscript. Below are some of my comments and suggestions to improve the article (these are examples):
- Page 1, line 37:Change double-strander to double-stranded. Randed to range.
Responds: We are appreciated for the reviewer’s comments. We had changed double-strander to doubled stranded. We sincerely apologize for this oversight and truly appreciate you pointing it out.
- Page 1:Provide a citation to direct readers interested in the mtDNA interactome for the sentence: “The human mitochondrial genome is a 16,569 bp double-strander circular DNA and is packaged by a ranged of proteins.”
Responds: Thanks for reviewer’s good suggestion. We have added references in our revised manuscripts (PMID: 22142616,PMID: 39500273). Please see page 2.
- Page 1, line 39:Replace distribution with content.
Responds: We had replaced distribution with content as reviewer’s suggestion.
- Page 2, line 54:Provide a more appropriate citation for MAVS to direct interested readers.
Responds: We appreciated the reviewer for pointing that out. We have added some foundational works references in our revised manuscripts (PMID: 16153868 ,PMID: 16125763, PMID: 16177806,PMID: 16127453). Please see page 2, line 54.
- Page 2, line 59: Rewrite the sentence:
“Upon mitochondrial are damaged, such as oxidative stress or viral invasion, can lead to the release of mtDNA via pathways like BAX/BAK macropores, mPTP channel, and mitochondrial-derived vesicles (MDVs), thereby activating the cGAS-STING signaling pathway.”
for better clarity and grammar.
Responds: We apologized for the poor language. We had rewritten this sentence. “When mitochondria are damaged by events like oxidative stress or viral infection, mitochondrial DNA (mtDNA) can escape through several mechanisms, including BAX/BAK macropores, the mPTP, or mitochondrial-derived vesicles (MDVs). This released mtDNA then activates the cGAS–STING signaling pathway. ”
- Page 2, line 64:Replace citation #14 with PMID: 39406969, as the cited article does not discuss VDAC1 in the release of mtRNAs.
Responds: Thanks for the reviewer point out this. We had replaced the citation with PMID: 38955468.
- Page 2, line 92:Define MTERF1.
Responds: We defined MTERF1 as Mitochondrial Transcription Termination Factor 1.
Overall: The article is well-structured and easy to read.
Reviewer 2 Report
Comments and Suggestions for Authors
This review is well writting. The topic is very interesting and it is well described along in the manuscript.
The figures from the manuscript are very good graphical schematics of the things described in the paper.
Something to revise is that the initials from some words, like pseudouridylation (Ψ) and Pseudouridine (Ψ), are the same. Do you refer to the action or the the substance?
Author Response
This review is well writting. The topic is very interesting and it is well described along in the manuscript.
The figures from the manuscript are very good graphical schematics of the things described in the paper.
Something to revise is that the initials from some words, like pseudouridylation (Ψ) and Pseudouridine (Ψ), are the same. Do you refer to the action or the substance?
Responds: We appreciated the reviewer for pointing that out. You are absolutely right-Ψ refers to Pseudouridine, not pseudouridylation. We sincerely apologize for this oversight and truly appreciate you pointing it out. Please see our new revised manuscripts on page 4.
Reviewer 3 Report
Comments and Suggestions for Authors
The manuscript provides a timely overview of mitochondrial RNA biology and its links to innate immunity. It covers transcription, RNA processing, degradation, and pathological implications in depth. Significant revisions are required to improve clarity, integration, and forward-looking insight.
Major comments:
1- Much of the review is descriptive, with long lists of factors and mechanisms. In my sense readers expect more integration: how do these processes fit together? Where are the controversies? Which models are debated? A critical evaluation would strengthen the paper.
2- Many recent papers are included, which is excellent, but foundational works are under-cited. For example, earlier studies on mitochondrial RNA turnover and immunogenicity should be acknowledged. In addition some statements lack references.
3- Add a third figure to illustrate different pathways of mtRNA release and immune sensing.
4- Section 6 raises important questions about RNA marks, release mechanisms, vesicular export but does not provide testable hypotheses or potential experimental strategies.
5- The manuscript focuses almost exclusively on mRNAs, tRNAs, and rRNAs. However, mitochondrial non-coding RNAs (lncRNAs, circRNAs, small RNAs) have emerging roles in immunity and could enrich the review.
Minor comments:
Many typos. I recommend a thorough language revision.
Ensure all abbreviations are defined upon first use.
Some references are redundant.
Comments on the Quality of English LanguageI recommend a thorough language revision.
Author Response
The manuscript provides a timely overview of mitochondrial RNA biology and its links to innate immunity. It covers transcription, RNA processing, degradation, and pathological implications in depth. Significant revisions are required to improve clarity, integration, and forward-looking insight.
Major comments:
- Much of the review is descriptive, with long lists of factors and mechanisms. In my sense readers expect more integration: how do these processes fit together? Where are the controversies? Which models are debated? A critical evaluation would strengthen the paper.
Responds: We sincerely thank the reviewer’s comments. In response, we have briefly described the relationship between mitochondrial RNA maturation and decay processes, as well as how abnormalities in these processes may lead to RNA accumulation. Additionally, in the section regarding the "tRNA punctuation model," we have discussed the challenges to this theory posed by mitochondrial non-coding RNAs (Please see page 4). We hope these additions will help readers achieve a better integration of the information presented.
- Many recent papers are included, which is excellent, but foundational works are under-cited. For example, earlier studies on mitochondrial RNA turnover and immunogenicity should be acknowledged. Inaddition some statements lack references.
Responds: We are appreciated for the reviewer’s suggestion. We have added citations to several key earlier studies on mitochondrial RNA turnover and immunogenicity, as suggested. Furthermore, we have revised the manuscript to ensure that all appropriate statements are supported by relevant references.
- Add a third figure to illustrate different pathways of mtRNA release and immune sensing.
Responds: Thank you for this excellent suggestion. We agree that a clear illustration of the pathways of mtRNA release and immune sensing would be valuable for the readers.
In our manuscript, we have actually consolidated this complex information into a comprehensive model presented in Figure 2. This figure was specifically designed to illustrate the known and potential pathways of mtRNA release (including via BAK/BAX pores, ANT2, VDAC1 channels, and potential routes like MDVs or the MLKL pore) and the subsequent activation of immune sensors (such as MAVS, PKR, and the proposed TLR3 pathway).
We have now revised the figure legend and the corresponding main text to describe these pathways more explicitly and to ensure that the figure is clearly interpreted as the central summary of this mechanism. We believe that Figure 2, in its current form, effectively serves the purpose of the suggested third figure by providing a complete and integrated overview of the current understanding of mtRNA release and immune activation.
4- Section 6 raises important questions about RNA marks, release mechanisms, vesicular export but does not provide testable hypotheses or potential experimental strategies.
Responds: We have added our hypotheses and experimental strategies as reviewer’s suggestion. Please see page 11.
5-The manuscript focuses almost exclusively on mRNAs, tRNAs, and rRNAs. However, mitochondrial non-coding RNAs (lncRNAs, circRNAs, small RNAs) have emerging roles in immunity and could enrich the review.
Responds: That is an excellence question. We completely agree that mitochondrial non-coding RNAs represent an emerging and crucial area in immune regulation.
Following your advice, we have expanded the manuscript to include a discussion on the functional roles of mitochondrial non-coding RNAs in immunue responses. This new content can be found in a dedicated subsection titled "5.3 Mitochondrial noncoding RNAs mediated immune responses" on pages 10 and 11.
We believe this important addition significantly enriches the review and provides a more comprehensive perspective. Thank you again for your thorough review and insightful comment.
Minor comments:
Many typos. I recommend a thorough language revision.
Ensure all abbreviations are defined upon first use.
Some references are redundant.
Responds: We apologize for the poor language for our manuscript. We worked on the manuscript for a long time and the removal of sentences and section obviously led to poor readability. We agree with this suggestion and have modified the details in the experiments and figures as appropriate. We're going to have native English speakers for language corrections. We really hope that the language level has been substantially improved.
Additionally, an abbreviations list has now been included (please see page 13), and redundant citations have been removed. We would like to express our thanks once again to the reviewers for highlighting these points.
Round 2
Reviewer 3 Report
Comments and Suggestions for Authors
No comments